# Regime-dependence when constraining a sea ice model with observations: lessons from a single-column perspective

Molly M. Wieringa<sup>1,2</sup> and Cecilia M. Bitz<sup>1</sup>

<sup>1</sup>University of Washington, Department of Atmospheric Sciences, Seattle, WA, USA

<sup>2</sup>NSF National Center for Atmospheric Research, Advanced Study Program, Boulder, CO, USA

Correspondence: Molly Wieringa (mollyw@ucar.edu)

Abstract. A substantial body of work has explored the use of sea ice concentration (SIC) and sea ice thickness (SIT) observations to initialize modeled estimates of the unobserved Arctic sea ice state via data assimilation (DA). While many recent studies have highlighted the particular value of incorporating SIT observations to this end, the influence of local sea ice conditions on the efficacy of assimilating various observation types has not been sufficiently evaluated. This work utilizes a single-column sea ice model to represent three common Arctic sea ice regimes: pack ice, seasonal ice, and first-year ice. An ensemble data assimilation framework is then used to assimilate synthetic observations of SIC, SIT, and two types of sea ice freeboard in each regime. Results demonstrate substantial variation in observation efficacy across observation types and sea ice conditions. In particular, SIT and laser altimeter freeboard observations are found to have a broadly positive impact in thick ice regimes, while SIC effectively constrains thinner, more marginal sea ice regimes. A need for regime-tailored DA strategies and further experimentation with underutilized sea ice observation types is strongly implied.

#### 1 Introduction

Constraining sea ice models with observations is critical for accurately estimating unobservable aspects of the sea ice system and for making reliable predictions of sea ice, weather, and climate conditions. Uncertainties in initial sea ice concentration (SIC), sea ice thickness (SIT), and snow properties can propagate and become persistent model biases. Data assimilation (DA) systems aim to mitigate these initial condition errors by integrating observations into model estimates of the state, but current sea ice observing systems and sea ice models present unique challenges. The relative efficacy of assimilating available observation types remains insufficiently quantified, and the processes by which new observation types are evaluated in existing DA systems can be expensive and difficult to interpret. Additionally, many previous sea ice DA studies have focused on pan-Arctic assimilation with complex DA schemes, masking regime-dependent DA performance in regions such as the multiyear ice pack, the seasonal or marginal ice zone, and first-year ice edge, all of which are characterized by unique thermodynamic and dynamic processes (Bitz & Roe, 2004; Maslanik et al., 2011; Årthun et al., 2012; Allard et al., 2018).

Using a recently introduced sea ice single-column ensemble data assimilation framework (CICE-SCM-DART; Wieringa et al. 2024, Riedel, Wieringa, & Anderson 2025), this study evaluates how assimilating observations of SIC, SIT, and two types of sea ice freeboard—derived from radar and laser altimetry technologies—impacts the analysis of sea ice states across three

distinct sea ice regimes. The CICE-SCM-DART framework provides an ideal testbed for isolating the DA impact on sea ice initial conditions. By simulating three single-column sea ice regimes—including (1) thick, deformed ice with substantial snow accumulation; (2) thinner, seasonal ice that experiences rapid growth and melt cycles; and (3) very thin, first-year sea ice that must re-form after each melt season—the impact of assimilating observations is efficiently localized to a single-grid cell and the effects of large-scale advection are excluded. The lightweight nature of CICE-SCM-DART enables rapid experimentation in each regime and allows for a systematic isolation of the contributions of observation type and regional ice processes to DA performance.

The observation types selected for this study are either already commonly assimilated (SIC), have been widely tested but are subject to practical limitations (SIT), or are underexplored but present promising avenues for sea ice initialization (freeboard). Passive microwave SIC observations provide broad spatial and high-frequency temporal coverage and have been widely assimilated into various sea ice and forecasting models (e.g. Lisæter et al. 2003; Schweiger et al. 2011; Posey et al. 2015), but they lack vertical ice information and do not effectively constrain estimates of sea ice thickness (Zhang et al., 2018). On the other hand, SIT observations have been shown to positively impact both sea ice volume and sea ice coverage estimates when assimilated (Lindsay & Zhang, 2006; Sakov et al., 2012; Yang et al., 2014; Ricker et al., 2017; Chen et al., 2017; Allard et al., 2018; Blockley & Peterson, 2018; Mu et al., 2018a, b; Xie et al., 2018; Zhang et al., 2018; Fritzner et al., 2019; Yang et al., 2020; Balan-Sarojini et al., 2021; Fiedler et al., 2022; Mignac et al., 2022; Cheng et al., 2023; Min et al., 2023; Williams et al., 2023; Zhang et al., 2023; Riedel & Anderson, 2024). In practice, SIT estimates are prone to temporal and spatial availability issues; additionally, remotely-sensed SIT observations are typically retrieved from altimetry measurements of sea ice freeboard, a process which requires ancillary snow data (Kwok & Cunningham, 2015; Petty et al., 2023) and introduces uncertainties that can limit the efficacy of the DA process (Petty et al., 2023). Conversion from freeboard to SIT also depends on the type of altimeter sensor used to take measurements. Radar altimetry (e.g. CryoSat-2; Kurtz & Harbeck 2017) penetrates snow layers to make measurements of the height of the snow-ice interface, whereas laser altimeters (e.g. ICESat-2; Kwok et al. 2023) capture the total freeboard height, including the snow atop the sea ice. Both radar freeboard (FBR) and laser freeboard (FBL) measurements are less uncertain than SIT measurements derived from them. Due to the seasonal cycle of snow accumulation, however, FBR and FBL can differ substantially and are not necessarily synchronized to the SIT seasonal cycle or each other (Fig. 2). Though a few studies have explored freeboard assimilation (Sievers et al., 2023; Mathiot et al., 2012), the comparative impact of these two types of freeboard observation is generally absent from existing literature.

Using a perfect-model approach, this study addresses three key questions:

- 1. How does assimilating SIC, SIT, FBR, or FBL observation impact the analyzed sea ice state in each of three characteristic sea ice regimes?
- 2. Do radar and laser freeboard assimilation yield divergent estimates of sea ice thickness and snow depth due to their sensor-specific measurement approaches?
  - 3. Which observation types most effectively constrain unobserved sea ice model state variables in each sea ice regime?

Though more realistic experimentation will be needed, this work provides substantive insights and examples for those seeking to optimize initial conditions for sea ice forecasting and state estimation in the polar regions. The paper is organized as follows: the methodology and experimental design are laid out in Section 2; results are presented in Section 3; a discussion of the results and implications constitute Section 4; Section 5 concludes.

#### 2 Methods

Experiments are performed using the CICE-SCM-DART framework (Wieringa et al., 2024; Riedel, Wieringa, & Anderson, 2025), which couples the Data Assimilation Research Testbed (DART; Anderson et al. 2009) to Icepack (Hunke et al., 2024b), the column-physics package of the Community Ice CodE (CICE; Hunke et al. 2024a) sea ice model. Icepack and DART are reviewed individually in Sections 2.1 and 2.2, respectively.

## 2.1 Icepack

80

To investigate the local impact of various sea ice observations on sea ice forecasting, Icepack is deployed as a single-column representation of the sea ice state. Like CICE, Icepack represents sea ice in each grid cell using a probability density function. Commonly referred to as the ice thickness distribution (ITD), this function describes the probability that sea ice has a particular thickness within the grid cell. Resolving the ITD allows the model to represent a range of thicknesses in each grid cell. This improves the model evolution of sea ice growth and melt, which are strong functions of sea ice thickness (Thorndike et al., 1975). Practically, the ITD resolves many sea ice variables in a discrete set of thickness categories.

The primary sea ice state variables in Icepack—sea ice area  $(A_{ice,n})$ , sea ice volume  $(V_{ice,n})$ , and snow volume  $(V_{sno,n})$ —have been discretized along the ITD and are hereafter referred to as "categorized" variables, where n indicates the category number. Icepack simulations used in this study are configured with 5 ITD categories, 3 snow layers, and 8 internal ice layers, as well as a mushy-layer thermodynamics scheme (Turner, Hunke, & Bitz, 2013) and a delta-Eddington shortwave radiation scheme (Briegleb & Light, 2007; Holland et al., 2012). The ITD categories are adjusted in response to thermodynamic and dynamic evolution using a linear remapping approach outlined by Lipscomb (2001).

While Icepack does not represent sea ice advection or motion in this single-column framework, it does include some representation of sea ice ridging. The amount of ridging is calculated from prescribed climatological rates of sea ice opening and closing rates; these data are available from observations taken during the SHEBA field campaign (Lindsay, 2002). During the ridging calculation, SIC is initially decreased as the ice converges. In earlier versions of Icepack, the sea ice contraction was assumed to increase the open water fraction. Sustained production of open water in a grid cell is not typically the case in CICE, however, as advective processes tend to replace the sea ice area fraction lost to ridging in the larger model. More recent versions of Icepack allow the user to address this discrepancy by instead choosing to replace the contracted SIC by sea ice of the same thickness distribution (Hunke et al., 2023). For this study, this newer option is selected.

For each experiment performed in this study, Icepack is used to generate a 30-member sea ice ensemble. Sea ice evolution in each member is forced by surface atmospheric variables from a localized version of atmospheric forcing data derived

from the Japanese 55-year reanalysis (JRA55-do, see *Prior Ensemble* for more details; Tsujino et al. 2018). Each ensemble member is also coupled to a slab ocean. While the atmospheric forcings supplied to each ensemble member vary by small perturbations, the ocean initial conditions and heat flux convergence forcing are identical for all ensemble members and are derived from the ocean component output of a fully-coupled historical simulation of the Community Earth System Model (CESM2; Danabasoglu et al. 2020).

#### 95 2.2 DART

100

DART is a modular ensemble DA program developed by the Data Assimilation Research Section at the NSF National Center for Atmospheric Research (Anderson et al., 2009). DART has been used extensively for research purposes and can be deployed into models of wide-ranging complexity and scale. A range of algorithmic options allows a user to flexibly tune the assimilation system, including a non-Gaussian filtering framework, 5 inflation approaches and several options for localization in large models.

In this study, DART is configured to assimilate observations using a non-Gaussian bounded rank histogram filter (Anderson, 2023; Wieringa et al., 2024; Riedel, Wieringa, & Anderson, 2025) and multiplicative inflation (Anderson & Anderson, 1999). The localization radius is set to infinity, as Icepack represents a single, isolated location. Synthetically derived observations of SIC, SIT, FBR, and FBL are assimilated into the model at daily timesteps. The weighted difference between the observation and the model estimate is then used to update the model's categorized state variables ( $A_{ice,n}$ ,  $V_{ice,n}$ , and  $V_{sno,n}$ ).

In contrast to the categorized structure of the sea ice model, observations of sea ice area and sea ice thickness are typically not probabilistic, but are instead point measurements of a single quantity. Therefore, assimilating realistic observations of the sea ice state requires implementing a translation, or forward operator, from the model's categorized variables to a modeled estimate of the observed quantity. These forward operators are defined in DART for each type of synthetic observation. For example, the SIT forward operator relates the categorized state to SIT as

$$SIT = \sum_{n=1}^{5} \frac{V_{ice,n}}{A_{ice,n}}.$$
(1)

Similar expressions are defined in DART for SIC,

$$SIC = \sum_{n=1}^{5} A_{ice,n},\tag{2}$$

and each of the freeboard observation types,

115 
$$FBR = \sum_{n=1}^{5} V_{ice,n} \times (1 - \frac{\rho_i}{\rho_w}) - V_{sno,n} \times (\frac{\rho_s}{\rho_w}),$$
 (3)

and

$$FBL = \sum_{n=1}^{5} V_{ice,n} \times \left(1 - \frac{\rho_i}{\rho_w}\right) - V_{sno,n} \times \left(\frac{\rho_s}{\rho_w} - 1\right),\tag{4}$$

where  $\rho_i = 917.0kg/m^3$ ,  $\rho_s = 330.0kg/m^3$ , and  $\rho_w = 1026.0kg/m^3$  are assumed values for the density of ice, snow, and sea water, respectively. Note that the only difference between FBR and FBL from the model's perspective is the addition of snow depth in laser altimeter freeboard estimates. Since the forward operators tend to generate observation estimates by summing across the categorized state variables, these quantities (e.g. SIT, SIC, FBL, and FBR) are hereafter referred to as "aggregate" variables.

To ensure physical continuity, any nonphysical modeled values of SIC or SIT after the assimilation completes are post-processed to ensure that they do not violate physical bounds on SIC ([0, 1]) or SIT ( $[0, \infty]$ ) or the requirements of monotonically increasing midpoint thickness in the model's categorized ice thickness distribution. When using the non-Gaussian filtering framework in DART, much of the need for postprocessing is eliminated. However, the categorized nature of the sea ice state variables can lead to inappropriate values when attempting to constrain both sea ice area categories and total SIC (Wieringa et al., 2024; Grooms & Riedel, 2024). Therefore, if the analysis SIC produced by the DA filter is greater than 1.0, the postprocessing rescales the individual ice area categories in the model as

$$130 \quad A_{ice,n}^{pp} = A_{ice,n}^a \times \frac{1}{SIC^a}, \tag{5}$$

where superscript pp indicates the postprocessed values and superscript a indicates the analysis values produced by the DA filter. To ensure basic physical consistency, the ice volumes in each category are recalculated using category midpoint thicknesses,

$$V_{ice,n}^{pp} = A_{ice,n}^{pp} \times h_{mid,n}. \tag{6}$$

This ensures that individual category thicknesses  $(v_{ice,n}/a_{ice,n})$  will not (a) violate the assumption of monotonically increasing thickness in the ITD; nor (b) violate the thickness bounds of the categories themselves.

As a final step, the postprocessing addresses cases in which the assimilation produced ice where there was no ice in the prior. Volume in each category for which this occurs is calculated according to Equation 6. The column layer ice enthalpies and salinites are initialized as though the ice was new, following extant conventions in Icepack. Similar conventions are followed to initialize snow layer enthalpies if the assimilation produces snow where there previously was none. If the assimilation removes all the ice area from a category, the ice and snow volumes, as well as the column enthalphies and salinities, are all set to 0.0.

## 2.3 Experimental Setup

120

125

145

The perfect-model experimental approach adopted in this study reduces the complexity of the sea ice DA problem by eliminating a few contributions to observational and model uncertainty. The assimilated observations are drawn from a randomly-selected member of a prior, unassimilated sea ice model ensemble, nullifying the need to consider model-observation mismatches attributable to imperfect model physics. In this work, the synthetically-derived observations are also assumed to be perfectly representative on the model grid, thereby eliminating scale-related uncertainties. These conditions simplify the DA implementation, but require that the results, as far as observation impact on the sea ice state, be interpreted as a "best-case scenario."

Each perfect model experiment uses CICE-SCM-DART to assimilate daily sea ice observations into a prior sea ice model ensemble. The prior ensembles have been configured to represent three different sets of sea ice conditions ("regimes"); synthetic observations are derived for SIT, SIC, FBR, and FBL and assigned an observational uncertainty estimate. The subsequent sections describe model ensemble configuration, observation derivations, and evaluative metrics in further detail.

## 2.3.1 Prior Ensembles

Ensemble DA requires a prior ensemble estimate of the sea ice state, the variance of which provides a measure of prior uncertainty. As this study is partially concerned with the impact of DA across sea ice regime types, three prior ensembles are considered (Table 1). The prior ensemble locations, each of which represents an Arctic grid cell, are selected as representative of various sea ice conditions that exist in the Northern Hemisphere; they are subsequently referred to as PACK ICE, SEASONAL ICE, and FIRST-YEAR ICE. Delineation between the three regime types is based on the annual cycle of sea ice concentration. PACK ICE conditions are considered to be those for which ensemble mean SIC remains above 0.8 throughout the year. The SEASONAL ICE regime is then defined by ice locations that experience periods of ensemble mean SIC greater than 0.8 and less than 0.15, and FIRST-YEAR ICE is that for which all ensemble members melted away completely during at least some portion of the year. Based on these criteria, three locations in the Arctic Ocean were selected to represent the respective regimes in this work. Icepack itself is agnostic as to the location it is representing—the atmosphere and ocean forcings produce sea ice conditions representative of the intended regime. To generate an ensemble of sea ice states at each location, 30 versions of global atmospheric conditions are perturbed from the JRA55-do by adding small amounts of noise to the dominant patterns of interannual variability (see Appendix A). From this perturbed atmosphere ensemble, the local surface conditions at each location are extracted and rewritten to input files for Icepack. Local initial conditions for each ensemble's slab ocean model were similarly extracted at each location but are identical across each ensemble's individual members. The specific locations 170 used to isolate these forcings are illustrated in Fig. 1.

Each of the three ensembles is spun up for a 10-year period from atmospheric forcing year 2000 through 2010. While this process allows the mean sea ice state to reach a relative equilibrium, it also crucially allows the sea ice simulations to diverge across the ensemble in response to perturbed atmospheric forcing. A 5-year ensemble for each location is then produced using restarts from the end of each spin-up run and atmospheric forcings from 2011 to 2016. These simulations are known as the FREE runs and serve as control experiments at each location (Fig. 3). Neither the spin-up nor the FREE ensembles undergo any assimilation.

#### 2.3.2 Observations


In a perfect-model experiment, observations are synthetically derived from a randomly-selected member of the FREE ensemble, hereafter referred to as the TRUTH. During the DA process, the TRUTH member is then subsequently withheld from the ensemble. To ensure that the impact of DA on each sea ice regime is compared across sea ice conditions that reflect the same Arctic atmosphere, the TRUTH member for each regime is forced by atmospheric conditions that come from the same randomly perturbed version of the JRA55-do. In each regime, a prescribed observation error variance value is added to generate a

distribution around the value of TRUTH at the desired time of observation. SIT, SIC, FBR, and FBL observations are randomly drawn from this distribution around the TRUTH ensemble member. The end result is a set of observations that are intentionally "noisy" to within expected observational errors (Fig. 4).

Observational errors are vastly simplified when prescribed in the perfect-model framework. Because the observation comes from the model itself, various kinds of representativeness error are avoided, including bias and the impact of missing physics. In this context, the error values can be thought of as largely indicative of instrument uncertainty and observational pre-processing. The prescribed errors for each observation type used in this work are listed in Table 2. For a thorough explanation of the estimation of observational errors for these perfect-model experiments, please see Appendix B.

Observations are extracted from TRUTH every 24 hours over the course of atmospheric forcing year 2011. In all experiments, the TRUTH is used as the validation data.

#### 2.4 Evaluative Metrics



To quantify the impact of assimilating daily observations in the reconstruction experiments, the mean absolute error (MAE) between the ensemble mean of each experiment (EXP) and TRUTH is calculated. MAE, which is calculated as,

$$MAE = \sum_{i}^{n} \frac{|EXP_i - TRUTH_i|}{n},\tag{7}$$

represents the mean difference between the analysis and the TRUTH over the course of the reconstruction period. Because the values of SIC and SIT vary across the PACK ICE, SEASONAL ICE, and FIRST-YEAR ICE ensembles, it might be expected that the magnitude of adjustments (and therefore MAE) might also vary quite a bit. To generalize this metric for easy comparison of assimilation impact across regimes, MAE is translated into a percent MAE reduction (pMAE) by normalizing each experiment's MAE by the FREE MAE,

$$pMAE = 100 \times \frac{MAE_{FREE} - MAE_{EXP}}{MAE_{FREE}}.$$
(8)

The results can then be interpreted outside the context of mean state differences, and more easily compared in terms of assimilation impact.

Statistically significant differences between reconstruction and initialization experiments, the FREE ensemble, and TRUTH are determined using a Welch's t-test. In this study, significance should be interpreted with care, as the nature of perfect model experiments may necessitate assimilating observations from a TRUTH that is not itself significantly different from the FREE ensemble mean.

## 3 Results

Figures 3 and 5 demonstrate the ability of Icepack to capture the chosen Arctic sea ice regimes. In the pack ice regime (PACK ICE), which is forced by atmospheric conditions at 88N and 0E, SIC is always close to 1 and SIT is relatively steady at 3–4m (Fig. 3, panels a,d). Snow depth (SND) exhibits a strong seasonal cycle, and can exceed 70cm near the boreal spring sea ice

maximum (Fig. 3, panel g). As a consequence of the near-total sea ice coverage and summertime surface temperatures that hover near freezing (Fig. A1), the modeled ensemble spread in PACK ICE SIC is quite small (Fig. 3, panel a; Table 1).







The second regime, based on more southerly conditions in the Chukchi Sea (75.53N, 174.45E), is representative of a seasonal ice environment (SEASONAL ICE), in which sea ice retreats to low concentrations during the melt season and advances to complete coverage after freeze-up. In this regime, modeled ensemble spread for SIC, SIT and SND is relatively low (Fig. 3, panels b,e,h; Table 1), as the ensemble is constrained by atmospheric conditions that drive all ensemble members toward maximal and minimal sea ice coverage, with limited opportunity to diverge from one another in the shoulder seasons. This occurs without any change to the ensemble variance in atmospheric conditions, and is a feature of sea ice boundedness in a regime that exhibits both sustained freezing and melting conditions. Limited model ensemble spread in seasonal ice regimes should be an expected feature of sea ice data assimilation applications that requires special attention to address.

The third regime represents first-year ice conditions (FIRST-YEAR ICE) in the Barents Sea (75N, 40E), in which sea ice retreats completely during the melt season and advances as new growth each year. In contrast to the other two regimes, FIRST-YEAR ICE has comparatively large model ensemble spread in SIC (Fig. 3, panel c; Table 1), due to the sensitivity of new ice growth to the first date and persistence of freezing conditions at this forcing location. Though total sea ice volume in this ensemble is quite low, ensemble spread in SIT and SND is large compared to PACK ICE and SEASONAL ICE, due to the large spread in SIC (Fig. 3, panels f,i).

To examine how representative these ensembles are of ice conditions in a fully-dimensional thermodynamic-dynamic sea ice model, the model ensemble covariance relationship between SIC and SIT in each regime is compared to the covariance range in a CICE6 ensemble simulation forced with the same JRA55-do data atmosphere and ocean conditions (Fig. 5). In the left column, SIC variance, SIT variance, and their covariance over the 2011-2016 FREE run period are plotted as a function of mean thickness for all Arctic grid cells in the CICE6 simulation and for all members of each Icepack regime ensemble (panels a, c, e). In the right column, the daily variances and covariances across each Icepack ensemble (solid lines) are compared to the daily covariance of the corresponding grid cell from the CICE6 simulation (shaded regions) over the 5 year FREE period (panels b, d, f). In general, the single-column ensembles capture reasonable sea ice variance across their respective regimes, fitting in nicely to the distribution of Arctic grid cells and mirroring the timing of variance peaks and troughs at corresponding grid cell locations in CICE6. Though the PACK ICE grid cell appears to be a particularly low-variance selection (panels a, c), both daily SIC and SIT variance are underestimated in Icepack (Fig. 5, panels b and d), due at least in part to the absence of sea ice dynamics in the column model. In addition to a lack of sea ice advection (which would generate a sort of Eulerian variability), ridging also leads to increased SIT variability are under-represented in the Icepack ensembles. Despite this, the single-column model regimes mimic the ice conditions in a more complex model relatively well, lending credence to their use as testing ground for sea ice DA.

Results are subdivided into (a) the impact of assimilating each observation type in each location on the *model's estimates* of those same observable quantities; and (b) the observations' impact on the each category of the *model's underlying state* variables.

## 3.1 Impact on the aggregate state




Figure 6 (panels a, c, e) shows the ensemble mean sea ice volume result when assimilating SIC, SIT, FBR, and FBL observations, compared to the unassimilated FREE mean and the TRUTH ensemble member. While later sections will quantify the impact of assimilation on the observable variables themselves, aggregate sea ice volume provides an intelligible first perspective on how assimilating observations adjusts the sea ice state within a grid cell. In PACK ICE, SIT and freeboard observations tend to adjust the model's ice volume most toward the TRUTH, particularly in the latter half of the year, when ensemble spread in volume during this year increases. SIT observations demonstrate a particularly noticeable adjustment year round, which weakens only during the late summer months. Over the early part of the year, FBL observations have a negligible impact on sea ice volume, while FBR observations begin to have a positive impact in April. In contrast, SIC observations have a minimal to negative impact on recovering the PACK ICE TRUTH's ice volume at any time.

In SEASONAL ICE, the adjustments to sea ice volume are harder to evaluate visually, due to the narrow modeled ensemble spread in this regime. The consistent lack of spread is due to the highly seasonal nature of ice growth and melt in seasonal conditions. Because atmospheric conditions in this regime constrain nearly all ensemble members to full ice cover in the winter months, and strong feedbacks accelerate the loss of ice to near-zero in the summer months, ensemble spread is constrained to low values during these SIC extremes, limiting the period of appreciable ensemble spread to the freeze-up and melt seasons. What spread does arise during these shoulder seasons is largely the result of the timing with which each ensemble member begins freezing in fall and finishes melting the following summer. Despite the lack of spread, it can be noted that SIT observations again appear to reasonably recover the TRUTH for much of the year, with the exception of the early summer months. Freeboard and SIC observations again appear to do very little prior to July, but by the time freeze-up commences in the fall, all four observation kinds have adjusted the ensemble mean ice volume away from the unassimilated FREE case and toward TRUTH.

The primary efficacy of SIT observations for reconstructing sea ice volume declines slightly when examined in FIRST-YEAR ICE. SIT observations still more successfully capture the TRUTH than the FREE ensemble, but in this regime, where SIC ensemble spread is quite large due to the sensitivity of new ice to small differences in atmospheric conditions, SIC observations best recover the TRUTH throughout much of the year. The impact of freeboard observations differs notably in this regime; FBR observations appear to have a totally negligible impact on sea ice volume reconstruction in FIRST-YEAR ICE, while FBL observations recover sea ice volume comparably to SIC observations.

At first glance, there are few cases in which assimilating sea ice observations appears to positively impact aggregate snow volume (Fig. 6, panels b, d, f). In PACK ICE, FBL observations adjust the model toward TRUTH for a short period in the summer; by contrast, FBR observations degrade the model estimate of snow volume during this period. FBL observations also exhibit a positive influence on modeled snow volume in the thin FIRST-YEAR ICE regime, as do SIT and SIC observations, while FBR observations do little to adjust the model from the unassimilated FREE state. None of the assimilated variables demonstrate much impact on snow volume in SEASONAL ICE.

#### 3.1.1 Annual comparative error reduction






A comprehensive evaluation of annual assimilation impact on modeled observable variables is presented in Fig. 7. The percent MAE reductions for ice thickness largely mirror the results shown in terms of sea ice volume evolution in Fig. 6; in the PACK ICE and SEASONAL ICE cases, SIT observations have the largest impact on reconstructing thickness, followed by freeboard observations in PACK ICE and SIC observations in SEASONAL ICE. SIC observations, which have negative but insignificant impacts on all observable variables in PACK ICE, may constrain modeled SIC and SIT in SEASONAL ICE conditions, and have a significant positive impact on both modeled SIC and SIT in FIRST-YEAR ICE. Across all regimes, SIT observations provide the strongest constraint on modeled SIT. In thinner ice regimes, where model ensemble spread in SIC is greater, SIC observations most effectively constrain SIC conditions.

While none of the assimilated observation types have a significant impact on modeled annual SND, this is partly attributable to the fact that the TRUTH and FREE ensemble mean snow depths are not themselves significantly different from one another over large stretches of the year, especially in thicker ice regimes (not shown). The otherwise apparent inability of SIC and SIT observations to recover modeled snow depths aligns with a previous study that demonstrated the need for SND observations to appropriately constrain snow estimates in CICE5 (Riedel & Anderson, 2024). However, these experiments demonstrate that FBL observations may positively impact SIC and SND estimates in all regimes. While the magnitude of this impact varies, FBL observations tend to slightly improve SIC and at minimum avoid degrading SND estimates. On the other hand, FBR observations tend to improve estimates of SIT in thick ice cases, but may have a negative impact on modeled SND in across regimes. This is particularly true in the thick ice regimes, where thick ice allows for heavy snow loads; while the depth of snow increases, modeled FBR decreases as the snow-ice interface is depressed by the weight of the snow.

#### 3.1.2 Seasonal comparative error reduction

Because sea ice evolves seasonally, as does the sea ice ensemble spread, annual mean results may mask periods of greater or lesser impact. Figures 8 (winter, October-March) and 9 (summer, April-September) explore this idea further. The summer pMAE looks qualitatively the same as the annual relative error reductions, though the DA impact in summertime conditions tends to be more saturated and significant than in the annual perspective. Assimilating summertime observations thus largely dictates the sign of annual error reduction, which is then diluted by the inclusion of wintertime results. This is reasonable, given that ensemble spread tends to increase in the summer months for most variables, which enhances the influence of the observations in the DA adjustment process.

Wintertime relative error reductions are qualitatively consistent with annual pMAE reductions with a few notable exceptions (Fig. 8). In freezing conditions, modeled SIC estimates in PACK ICE are improved by assimilating any of the observation types. Conversely, modeled SIT estimates in SEASONAL ICE are degraded by all observation types. As implied by the lack of significance, improvements to SIC in PACK ICE are small in magnitude, given the negligible ensemble spread in SIC during this time. Negative impacts to modeled SIT in wintertime SEASONAL ICE can be attributed almost entirely to differences in timing of freeze-up date in this strongly seasonal regime. Declines in SIT occur rapidly after the onset of freezing conditions

in SEASONAL ICE, as ice area increases rapidly compare to ice volume (Fig.10). Small differences in the date of this rapid change from the TRUTH lead to large MAE values over a relatively short window of time, and an overall negative pMAE reduction. Over the rest of the winter period, the influence of DA on the modeled SIT in SEASONAL ICE is broadly positive (Fig.10).

## 3.2 Impact on the sea ice state

Since observable variables in Icepack are aggregates of the model's categorized variables, understanding the assimilation impact on SIT, SIC, and SND also requires understanding how assimilating observations impacts  $A_{ice,n}$ ,  $V_{ice,n}$ , and  $V_{sno,n}$ . The aggregation from model state variable to observable quantity occurs once in the beginning of the DA filtering process, to compare observations to model estimates of the observable quantities, and then again after the filter has been applied, to diagnose the adjusted modeled estimates of SIC, SIT, and SND. During the filtering process, the difference between the model estimates of observables and the observations themselves are regressed onto each category of Icepack's state variables using the model ensemble error covariance relationship between the observable quantities and the categorized variable in question. Any impact on an observable quantity is therefore a product of how the observations adjusted the categorized state variables.

In some cases for which modeled observables (e.g. SIC) are straightforward aggregations of the model's state variables  $(A_{ice,n})$ , the link between assimilation impact on state variables and modeled observables is more intuitive. In Figs. 11 and 12, assimilating SIC observations in PACK ICE has a negative impact on ice area and ice volume estimates in every thickness category; it is therefore no surprise that assimilating SIC observations has an overall negative impact on aggregated SIC and SIT in the PACK ICE regime (Fig. 7). However, improvement in individual categories does not automatically equate to improvement in the aggregate modeled observables. For example, in the same PACK ICE regime, assimilating SIT observations is found to significantly positively impact 3 of 5  $A_{ice,n}$  categories, yet has an insignificant negative impact on modeled SIC. Likewise, assimilating FBL observations produces a positive (though still insignificant) impact on modeled SIC, even though these observations also significantly positively impacted the same 3 of 5 ice area categories.

The most contrary results for sea ice area and volume tend to occur in the PACK ICE regime, where the representation of sea ice on a grid and the categorized nature of the sea ice model clash during DA. In thick ice conditions, each categorized variable exhibits reasonable spread across the ensemble and contains a portion of the total ice area for large parts of the year. However, aggregate ice coverage in the PACK ICE regime is always close to one, resulting in quite narrow ensemble spread in SIC. The initial adjustments to modeled SIC during assimilation are therefore small, since the DA filter weights the adjustment process against the observation when ensemble spread is less than observational uncertainty. However, because the ensemble spread in SIC is very narrow compared to spread in the model's categorized state variables, these small adjustments to SIC can be regressed into more substantial and significant adjustments to the categorized variables (i.e., adjustments to  $A_{ice,n}$  in PACK ICE). Since DA treats the categories of the state variables distinctly, there is no guarantee that each category of a given state variable is adjusted in a coordinated fashion within an ensemble member; improvements to individual categories could therefore combine into non-intuitive changes to aggregate variables.

In thinner regimes, ensemble spread in aggregate variables can be more proportional to ensemble spread in categorized variables. As SIC tends away from total sea ice coverage, the summation of the individual categories is less likely to always approach 1, reducing the cancellation of spread. However, in these regimes, ice is less evenly distributed across categories. For example, in SEASONAL ICE, most of the sea ice and snow resides in the thinnest 3 categories, while most of the FIRST-YEAR ice is contained in the thinnest category. Therefore, while assimilating aggregate observations like SIC does adjust the state in thinner ice cases, the majority of the impact on observable variables is attributable to the pMAE change in categories which contain the largest fraction of sea ice and snow in that regime.

Assimilating SIC or FBR observations tends to have a limited or negative impact on snow volume. In all three regimes, SIT and FBL observations do have a positive impact on at least some  $V_{sno,n}$  categories (Fig. 13); in the PACK ICE and SEASONAL ICE cases, however, the positive impact on snow volume occurs in categories that have less ice overall in them (the thinnest two ice categories in PACK ICE and the thickest two ice categories in SEASONAL ICE). This indicates that the positive impact on snow in these categories is negated by the negative impact on snow in the categories that make up more of the total snow volume in each regime. The exception to this pattern is the efficacy of SIC and FBL observations to improve modeled SND and  $V_{sno,n}$  estimates in FIRST-YEAR ICE, where ice is dominated by the thinnest ice category and ensemble spread is large during ice-covered portions of the year. Due to the large ensemble spread, adjustments are likely to be more accurate in the observable space and have a more reasonable covariance relationship with the state variables. Additionally, if the adjustment from an observation onto snow in the thinnest model category is positive, the overall adjustment to modeled SND is also likely to be positive, since the adjustments to the categories with very little ice or snow will contribute minimally to the overall impact.

#### 4 Discussion and Implications






This study has explored how assimilating plausibly available sea ice observations can impact the simulated sea ice state in various sea ice regimes. The single-column modeling framework is purposefully simplified, enabling rapid experimentation but obligating careful interpretation. The absence of coupling to sea ice dynamics and other components of the Earth system gives rise to important caveats that might be expected to lessen the impact of assimilated sea ice observations in fully coupled, three-dimensional experiments. The influence of advection and coupled feedback processes, which tend to reduce the persistence of sea ice anomalies, would need to be quantified in more comprehensive experiments. Using synthetic, perfect-model observations also reduces the demand for uncertainty quantification but sidelines questions related to observation rejection, representativeness, and bias. For these reasons, the results of this study should be considered an upper bound on the efficacy of assimilating sea ice observations. However, when appropriately contextualized, they offer valuable insights for future attempts to constrain the modeled sea ice state.

Across the three sea ice regimes, assimilating observations during the summer and early autumn freeze-up period demonstrated the greatest ability to reconstruct the true sea ice state. Of the observation types tested, SIT observations had the broadest, and often the largest, positive impact. Not only is MAE for modeled SIT reduced in all regimes, but assimilating

SIT also reduces modeled SIC error in seasonally ice-covered regimes. This result arrives at a promising time, as advances in observation processing increasingly enable year-round estimates of SIT from remote sensing observations (Landy et al., 2022). While real, basin-wide SIT observations are still relatively limited during the summer months, these findings should reinforce efforts to improve SIT observing systems during the summer, particularly preceding the transition to freezing conditions.

A novel finding of this study is the comparative impact of FBR and FBL observations, which differ in their impact on modeled snow depth in thick ice environments. While these two types of freeboard observations both appear to constrain thin ice environments very well, it must be noted that for present-day observing systems, both FBR and FBL observations are only available in regions that exceed a relatively high sea ice coverage threshold (Petty et al., 2023; Kurtz, Galin & Studinger, 2014). Thus, FBL observations are likely to have a more positive overall impact compared to FBR observations, as FBL may also improve SIC and avoid degrading SND in thick ice regimes. Additionally, while freeboard observations of either type display very limited impact in seasonal ice, SIC observations perform comparably to SIT observations in this regime. Given their year-round availability and relatively low observational uncertainty compared to current SIT observational estimates, assimilating a year-round combination of FBL and SIC observations is likely to produce the most accurate sea ice state estimate in current three-dimensional applications.

The impact of each observation kind, as well as the tendency for observations to be more effective during summer months, can largely be explained by the covariance relationships between variables across each of the sea ice ensembles. Figure 14 demonstrates how the model ensemble covariance relationship between observation types and total sea ice volume ( $V_{ice}$ ) evolves annually across sea ice regimes. In conditions where the ensemble spread of SIC is very small (PACK ICE, or SEA-SONAL ICE in the winter months), the covariance relationship between SIC and  $V_{ice}$  will also be very small, regardless of the comparatively large ensemble spread in SIT. The ability of SIC observations to update the model state is effectively quashed, as any small adjustment produced by comparing observed and modeled SIC values will be projected on the sea ice state variables via a very weak error covariance relationship. By contrast, in summertime SEASONAL ICE regimes, or FIRST-YEAR ICE, the ensemble spread in SIC is still seasonal but sufficient enough to allow SIC to adjust the model state via non-negligible summertime error covariances. While the details of these covariance relationships are dependent on how the model represents variability in the sea ice state, their general structure and seasonal evolution reflect physics that should be consistent across credible sea ice models available today.

Relative magnitude of observational error to model ensemble spread also contributes to the apparent potency of observation types. For example, SIC observations are particularly effective in FIRST-YEAR ICE in part because SIC observational error is prescribed as a parabolic function of SIC value. In the context of low observational uncertainty in a low-SIC environment (Fig. 4, panel c), the DA is heavily weighted toward the observation, because ensemble spread in SIC is large by comparison. A similar argument helps explain the switch in relative impact of FBR and FBL observations between thick and thin ice regimes. The observational errors assigned to synthetic FBL observations are selected as a positive linear function of FBL value (see Appendix B; Fig. 4, panels j-l). In thick ice environments, the ratio of FBL observational error to ensemble spread in modeled FBL will be lower than in thin ice environments. By comparison, the errors assigned to FBR observations are drawn from a

fixed range, regardless of FBR value (Fig. 4, panels g-i). The ratio of observational error to ensemble spread shifts in reverse and FBR observations become more effective for constraining modeled SIT in thick ice environments.

## 415 5 Conclusions





In the presented series of perfect-model, single-column sea ice modeling experiments, the ability of DA to constrain the sea ice state is found to vary as a function of sea ice observation type and sea ice regime. SIT and FBL observations are broadly the most effective across regimes, though SIC observations perform well in seasonal and very thin sea ice environments. If assimilated together in more realistic experiments, it is possible that SIC observations may compensate for the lack of real SIT and FBL observations in marginal ice conditions. Seasonally, observation impact tends to be larger in summer, which exhibits more substantial model ensemble spread across variables. In winter, error metrics for SIT in a single-column model are sensitive to the timing of freeze-up in very seasonal ice regimes, but observations otherwise broadly improve sea ice estimates during this time.

Simulating an ITD within a sea ice model complicates the assimilation process. Summarily, this work demonstrates that an improved estimate of an observable aggregate quantity does not guarantee an accurate adjustment of the underlying model state, which may influence the model's subsequent sea ice forecast. As many sea ice models employ an ITD, the relationship between observations and the model's categorized variables should be carefully evaluated.

Rapid hypothesis testing in CICE-SCM-DART enables a focused analysis of the impacts and performance of DA in a complex sea ice model. While there are notable caveats, the ease of simulation and interpretation in this single-column framework call attention to many intricacies and nuances relevant to the design of optimal sea ice data assimilation applications. Use of CICE-SCM-DART provides an accessible and necessary first step when developing and steering efforts to improve sea ice data assimilation.

Code and data availability. All code used in the study can be found on Github. The Icepack single column model is available from the CICE Consortium at https://github.com/CICE-Consortium/CICE. The Data Assimilation Research Testbed is maintained by DAReS and hosted at https://github.com/NCAR/DART. The version of DART used for this study was forked to https://github.com/mollymwieringa/DART. The postprocessed experiment data used to produce the figures is available upon request.

*Author contributions.* Both authors contributed to the conceptualization of the study. MMW performed the experiments, analyzed the results, and wrote the manuscript.

Competing interests. The authors declare no competing interests.

Acknowledgements. This material is based upon work supported by the National Center for Atmospheric Research, which is a major facility sponsored by the National Science Foundation under Cooperative Agreement No. 1755088. CMB and MMW acknowledge support from NASA ROSES Grant Number 80NSSC21K0745 and NSF Grant Number PLR-1936428. MMW was also supported in part by the University of Washington College of the Environment's Integral Environmental Big Data Research Fund. The authors would like to especially thank the DAReS team at NSF NCAR/CISL, as well as Jeffrey Anderson, Christopher Riedel, Alek Petty, and David Bailey for many insightful conversations and helpful technical support. We thank the editor and anonymous reviewers for constructive comments that helped improve the manuscript.

Figure 1. Selected locations for each sea ice regime type. The Arctic grid cells from which atmospheric conditions are drawn are shown for each experimental ensemble. The red dot represents the PACK ICE location at 88N, 0E; the blue dot is the SEASONAL ICE location at 75.53N, 174.45E; the white dot is the FIRST-YEAR ICE location at 75N, 40E. The annual mean sea ice concentration for a CICE6 ensemble during the same year as the single-column experiments is plotted in shades of white to blue (SIC of 1 to SIC of 0). The green contours represent annual mean SIC of > 0.8 (light green), > 0.5 (teal), and > 0.15 (navy blue).

Figure 2. Annual evolution of FBR, FBL, and SIT. The annual cycle of freeboard and sea ice thickness in various sea ice regimes is shown for the TRUTH state in atmospheric forcing year 2011. Model estimates of radar altimeter freeboard (FBR) are plotted as the height of the snow-ice interface (light blue shaded region) above 0.0, while laser altimeter freeboard (FBL) estimates are plotted as the height of the snow and ice surface (shaded ivory region) above 0.0. The total sea ice thickness is shown as the sum of the light blue and teal shaded regions. Maximum FBR (light blue circle), FBL (grey circle), and SIT (dashed blue line) are also plotted as illustrations of the seasonal timing of each observation type.

Figure 3. Icepack FREE ensemble regimes. The sea ice state in each FREE ensemble is presented for atmospheric forcing year 2011 in terms of SIC (top row), SIT (middle row), and SND (bottom). The annual evolution of each variable is presented in the dashed black line. The ensemble spread, calculated as the maximum and minimum value across the ensemble at each timestep, is plotted as the grey envelope. The individual ice thickness distribution categories for ice area ( $A_{ice,n}$ , top row), ice volume ( $V_{ice,n}$ , middle row), and snow volume ( $V_{sno,n}$ , bottom row) are plotted in similar fashion in the colored lines and envelopes. Note that though ice and snow volume are not the same metric as ice thickness and snow depth, the units in the single column model are equivalent for the purposes of plotting ( $m^3/m^2$  to m).

Figure 4. Synthetic observations extracted from a randomly selected member of the FREE ensembles. The observations, which are subsequently assimilated into the reconstruction and initialized hindcast experiments, are shown in grey lines for SIC (top row), SIT (second row), FBR (third row), and FBL (bottom row) for each modeled ice regime. The TRUTH from which the observations are generated is shown in the solid purple line, while the FREE ensemble mean is shown in the dashed black line. The observation error standard deviation  $(1\sigma)$  is shown as purple shading around TRUTH.

Figure 5. Icepack ensemble comparison to CICE6. Across-ensemble statistics in the three single-column sea ice regimes compared to a full CICE6 simulation forced by the same atmospheric conditions are shown. In the left column, the SIC variance in time at each grid cell in the CICE6 simulation for the period 2011-2016 is plotted in green against mean ice thickness at each grid cell in panel (a). SIT variance in time is plotted in blue (panel c), and the covariance between SIT and SIC at each grid cell is plotted in pink (panel e). The same quantities for each member of the regime ensembles are plotted for comparison, with PACK ICE ensemble members plotted in red triangles, SEASONAL ICE plotted in blue squares, and FIRST-YEAR ICE plotted in black crosses. The color gradient for each plot represents latitude in CICE6, with lighter shades of each color indicating more northerly locations. On the right, the variance across ensemble members for each Icepack ensemble is plotted in the solid lines for SIC (panel b), SIT (panel d), and their covariance (panel f), using the same color assignment as in the left column. The CICE6 variance or covariance at each day of year across the 2011-2016 time period is plotted in the shaded regions for the three grid cells corresponding to the locations of the Icepack ensembles.

**Figure 6. Observation influence on reconstructed sea ice volume.** The results of assimilating SIC (solid teal line), SIT (solid purple), FBR (solid light blue), or FBL observations (solid dark blue) are shown in terms of sea ice volume for PACK ICE (top panel), SEASONAL ICE (middle panel) and FIRST-YEAR ICE conditions (bottom panel). The black line represents the FREE case (without assimilation) and the thin red lines are the randomly selected TRUTH. For the results shown, thick lines are ensemble means and shading represents the ensemble standard deviation around the mean. Observations are assimilated at daily intervals throughout atmospheric forcing year 2011.

Figure 7. Annual bias reduction as a function of observation kind and sea ice conditions. Annual percent MAE reduction (pMAE) in observable variables relative to the FREE forecast as a result of assimilating various observation kinds (x-axis) in each of three sea ice regimes (PACK ICE, SEASONAL ICE, and FIRST-YEAR ICE, from left to right). Results are shown for modeled SIC (top row), SIT (second row), and SND (bottom row) and are calculated across all months of the year. Blue colors indicate a more beneficial impact due to assimilation, while red colors indicate a negative impact. The numbers indicate the specific pMAE associated with each experiment. Gray shading indicates that the analysis (EXP) is not significantly different from the FREE ensemble mean.

Figure 8. Boreal winter pMAE reduction in observables. Same as Fig. 7 but where pMAE is calculated over October-March.

Figure 9. Boreal summer pMAE reduction in observables. Same as Fig. 7 but where pMAE is calculated over April-September.

Figure 10. Observation influence on sea ice thickness in SEASONAL ICE. Same as Fig. 6 but where the y-axis represents SIT. Note the large single-day differences between each experiment (colored lines) and the TRUTH (red line) around the fall freeze-up event, when SIT declines rapidly as sea ice area expands.

Figure 11. Reduction in pMAE for categorized ice area state variables. Same as Fig. 7 but where pMAE has been calculated for each category of sea ice area  $(A_{ice,n})$  along the ice thickness distribution over all months of the year.

Figure 12. Reduction in pMAE for categorized ice volume state variables. Same as Fig. 11 but for categorized sea ice volume  $(V_{ice,n})$  along the ice thickness distribution.

Figure 13. Reduction in pMAE for categorized snow volume state variables. Same as Fig. 11 but for categorized snow volume  $(V_{sno,n})$  along the ice thickness distribution.

Figure 14. Model ensemble error covariance relationships across ice regimes. The ensemble error covariance of SIC (top row), SIT (middle row), and freeboard (bottom row) with total sea ice volume as function of time for PACK ICE (red), SEASONAL ICE (blue), and FIRST-YEAR ICE (black) regimes. The dotted lines in the bottom panel indicate FBL- $V_{ice}$  error covariances, while the solid lines show the FBR- $V_{ice}$  relationship.

**Table 1. Characteristics of prior ensembles.** The prior ensembles characterizing each of three sea ice regimes are outlined above, including location, annual mean SIC and SIT, and maximum ensemble spread in SIC and SIT over the assimilation period.

| Prior Ensemble | Location        | Annual Mean | Annual Mean | Max. Ens.    | Max Ens.     |
|----------------|-----------------|-------------|-------------|--------------|--------------|
|                |                 | SIC         | SIT         | Spread (SIC) | Spread (SIT) |
| PACK ICE       | 88N, 0E         | 0.971       | 3.396       | 0.036        | 0.378        |
| SEASONAL ICE   | 75.53N, 174.45E | 0.735       | 1.839       | 0.476        | 1.58         |
| FIRST-YEAR ICE | 75N, 40E        | 0.253       | 0.635       | 0.322        | 0.752        |

**Table 2. Observation error estimates as a function of observation kind.** Observation kind refers to the type of observation assimilated. Observation error refers to the formula used to determine an individual error estimate for each observation at each time step.

| Observation Kind | Observation Error                          |
|------------------|--------------------------------------------|
| SIC              | $\sigma_{SIC} = -0.5(SIC^2 - SIC)$         |
| SIT              | $\sigma_{SIT} = 0.1SIT$                    |
| FBR              | $\sigma_{FBR} \in [0.1, 0.15]$             |
| FBL              | $\sigma_{FBL} = 0.5FBL(1+x), x \in (-1,1)$ |

## Appendix A: Perturbed atmosphere method

Ensemble data assimilation requires an ensemble of model simulations that can maintain some spread among members, as a measure of uncertainty in the ensemble mean estimate of the model state. If ensemble spread collapses relative to the uncertainty of the observations being used to constrain the model, then the data assimilation will have no effect on the model state.

Several approaches to generate ensemble spread have been applied to sea ice data assimilation problems (Massonnet et al., 2015; Mu et al., 2018a; Zhang et al., 2018; Williams et al., 2023; Sievers et al., 2023; Riedel & Anderson, 2024). In this work, a perturbed atmosphere method is used, as it allows the sea ice model ensemble to develop spread that is related to variable atmospheric conditions, thus sampling sea ice conditions as a function of climate forcing. Currently, there exist some large ensemble atmosphere reanalyses that can provide variable atmospheric conditions to a sea ice model ensemble (e.g. CAM6-DART; Raeder et al. (2021)) without requiring any kind of manual perturbation, though they are not yet formally supported within CESM2.3. As such, the model simulations presented in this thesis are driven by a perturbed version of the single JRA55-do atmospheric forcing that is already actively supported by CESM2.3 as a data atmosphere model. The method outlined here was developed by François Massonnet and his original scripts are available upon request.

## A1 General Approach





As highlighted by Massonnet (*pers. comm.*), simple perturbation of the atmospheric fields in the forcing set (temperature, specific humidity, zonal and meriodional wind, short- and longwave flux, and precipitation) by the addition of white noise is untenable, due to the spatiotemporal coherence between these fields. Optimally, the perturbations produced from the reference forcing would maintain the statistics of the reference.

To generate sea ice conditions that reflect realistic atmospheric conditions, the desired perturbed atmospheric conditions must differ from one another, but maintain the spatiotemporal coherence between atmospheric variables. To accomplish this, the covariance matrix of the atmospheric state,

$$C = \frac{XX^T}{(n-1)},\tag{A1}$$

is preserved in the creation of each perturbation set by adding white noise to the left singular vectors of the singular value decomposition (SVD) of *C*. If the SVD is defined as

$$XX^T = U\Sigma\Sigma^T U^T, \tag{A2}$$

then a left singular matrix can be expressed as

$$R = \frac{U\Sigma^{1/2}}{(n-1)^{1/2}}.$$
(A3)

R can then be perturbed to Y via multiplication with a vector of random variables with zero mean and identity covariance,

$$Y = R \cdot z. \tag{A4}$$

The expectation of Y is shown to be 0,

$$\mathbf{E}(Y) = \mathbf{E}(R \cdot z) = R \cdot \mathbf{E}(z) = 0,$$
(A5)

while the covariance matrix of Y is equal to the covariance matrix of X,

$$\mathbf{480} \quad \mathbf{E}(YY^T) = Rzz^T R^T = R\mathbf{I}R^T = RR^T \tag{A6}$$

$$RR^T = \frac{U\Sigma\Sigma^T U^T}{(n-1)} = \frac{XX^T}{(n-1)} = C. \tag{A7}$$

In this framework, any number of arbitrary versions of z can be used to produce an equivalent number of perturbed atmospheric states, Y, that have statistics consistent with the original reference X.

To implement the general approach, additional data preparation and considerations for computational cost are necessary.

These are outlined in the next section.

## A2 Application

To obtain the perturbed versions of the JRA55-do surface variables produced by Tsujino et al. (2018), the general approach is divided into three processes: (1) preprocessing, (2) perturbation calculation, and (3) postprocessing.

In Step 1, daily means of the atmospheric fields are averaged from their native timestep (in the case of JRA55-do, 3-hourly). Leap days are removed, should they exist, and the year-to-year differences in each variable at each location in the dataset are calculated. In step 2, the data will be centered by removing the mean values of a reference period, so the range of data for which year-to-year differences need to be calculated must also include the full reference period (i.e. to obtain perturbations for 1980-1990 based on anomalies from reference period 1975-1985, preprocessing must include years 1975-1990).

Step 2 defines the perturbations using the preprocessed data defined in Step 1. First, for each variable of interest in the forcing dataset, the data for each timestep in the reference period is reshaped into a vector and stored in a reference period matrix (X'), from which a mean state vector is calculated. The anomalies for the reference period are then determined by removing the mean state vector  $(\overline{X})$  from the reference period matrix. The anomalies can be shown to be equivalent to a reduced version of R, given that X represents state anomalies,

$$X = X' - \overline{X} \tag{A8}$$

The covariance matrix of X can be expressed as an SVD,

$$XX^{T} = \frac{U\Sigma\Sigma^{T}U^{T}}{(n-1)},\tag{A9}$$

meaning that X, the state anomalies matrix, can also be defined as

$$X = \frac{U\Sigma}{(n-1)^{1/2}},\tag{A10}$$

which is equivalent to the definition of R introduced in the general application. Recognizing that for an nxm anomaly matrix X defined by n timesteps there are only n nonzero entries in  $\Sigma$ , R can more practically taken as the product of the n left singular vectors  $(U_n)$  and a reduced nxn version of the eigenvalue matrix (S),

$$R = \frac{U_n S^{1/2}}{(n-1)^{1/2}}.$$
(A11)

This reduces the overall computational load by removing the need to actually calculate the SVD of the covariance matrix.

Instead, the desired number of perturbations (p) for each year of the chosen forcing period can be straightforwardly produced by multiplying the reference period anomalies of each variable by a set of p random vectors, z.

Finally, in Step 3, once the perturbations have been calculated, they are interpolated back to the reference dataset's native timestep (3-hourly) and added to the original forcing values to produce p versions of perturbed atmospheric conditions. For additional flexibility, a multiplicative factor  $\alpha$  is used to control the amount of perturbation added to the original forcing. If the desired perturbation is less than might be observed due to interannual variability,  $\alpha < 1$ ; if the desired perturbation should exceed interannual variability,  $\alpha > 1$ . To generate ensemble spread that reflects year-to-year variations in daily sea ice estimates, the perturbations are applied using  $\alpha = 1$ .

## JRA55-do Localized Arctic Forcings

**Figure A1. JRA55-do localized atmospheric forcings.** The perturbed atmospheric forcings in 2011 extracted from the JRA55-do reanalysis for ice-ocean models are shown for key variables. The 30-member perturbed ensemble mean for 10m surface temperature (top); downwelling shortwave flux (second from top); downwelling longwave flux (second from bottom); and 10m specific humidity (bottom) are plotted for each of three locations in the Arctic Basin. The PACK ICE location (88N, 0E) is shown in purple, SEASONAL ICE (74.53N, 174.45E) in blue, and FIRST-YEAR ICE (75N, 40E) in teal. The grey shading in each panel represents the melt season, during which the PACK ICE location is consistently experiencing above-freezing temperatures.

## **Appendix B: Perfect-model observation error derivations**








In this work, a series of perfect-model Observing System Simulation Experiments (OSSEs) are formulated using CICE-SCM-DART and synthetically-derived observations. The use of such synthetic observations to explore various aspects of a data assimilation system or a specific type of observing network is common; in fact, DART provides ready-made programs to calculate synthetic observations from a randomly selected member of the ensemble forecast system. A DART user points such a program toward the randomly selected ensemble member and provides the location and an estimate of the error variance associated with each desired observation.

Sea ice observational products tend to vary widely in terms of the level of detail provided regarding observational error. For some products, observational error is given as a single value that approximates bias and precision (Zhang et al., 2018), while for others, each observation is associated with an instrument and algorithm error estimate (Kwok et al., 2023; Petty et al., 2023). It may also be the case that the OSSE experiment is being used to explore observation kinds that are not yet observed in the real-world, and for which observational error variance is a completely open question. When prescribing observational error variance in the course of producing synthetic observations in CICE-SCM-DART, there is therefore a wide range of approaches that could be taken for any given sea ice observation kind.

In this work, it is assumed that observational error varies in each kind of sea ice observation as a function of the the value of the observation itself. This can be the case for SIT and freeboard measurements, as error estimates associated with SIT and FBL observations from ICESat-2 can be generally noted to increase with ice thickness (Petty et al. (2023), Fig. B1). Based on a sampling of estimated linear relationships between ICESat-2 along-track FBL measurements and their associated uncertainty estimates, synthetic FBL error variance is assumed to be a quasi-randomized linear function of FBL (see Table 2). Uncertainties associated with ICESat-2 SIT estimates are much less linear, due to density assumptions and snow estimates used in the derivation of SIT from ICESat-2 FBL estimates. However, given the early stage of and somewhat arbitrary decisions involved in estimates ICESat-2 SIT, synthetic SIT error variance in this work is also assumed to be a linear function of SIT value (Table 2). The choice of scaling accounts for the general observation that FBL estimates tend to be 1/10th of associated SIT (Alexandrov et al., 2010; Sievers et al., 2023).

Radar freeboard error estimates are based on boreal wintertime measurements collected by the CryoSat-2 satellite (Fig. B1). The errors associated with CryoSat-2 freeboards fall between a minimum of 0.1m and 0.15m, regardless of the observed value or the time of year. To contextualize the results of this study in terms of realistic measurement systems, the FBR uncertainties prescibed here are randomly drawn from the range [0.1, 0.15] (Table 2).

Finally, for SIC observations, it is assumed that observation uncertainty is low when sea ice concentrations are very low or very close to total sea ice coverage (e.g. it is very obvious that there is or isn't ice covering the grid cell), but that the uncertainty increases for SIC values that represent more mixed divisions between ice cover and open water. To approximate this assumption, SIC observation error variance is taken to be an inverse parabolic function of the value of SIC itself, where error variance is near zero for SIC values at 0 or 1 and maximized for SIC near 0.5 (Table 2). The function has been scaled such

that the annual mean SIC error variance is near 15%, as has been used in previous OSSE studies (Zhang et al., 2018; Riedel & Anderson, 2024).

Figure B1. Freeboard error estimates based on observed freeboard value. The basis of prescribed observational error estimates for FBR and FBL observation types are illustrated using ICESat-2 observations (left; FBL observation type) and CryoSat-2 observations (right; FBR observation type). The real observation uncertainties provided by each satellite mission are plotted as a function of observation value for all measurements taken in 2011 (CryoSat-2) or 2019 (ICESat-2). Colors indicate the time of year each observation was taken. The red lines indicate the minimum and maximum uncertainties in the Cryosat-2 record and the dashed black line indicates the approximation used to determine the mean uncertainty for each observation value. The gray shading indicates the  $1-\sigma$  threshold random sampling of uncertainties around the mean value.

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
