# Peer review of "Regime-dependence when constraining a sea ice model with observations: lessons from a single-column perspective"

_EGUsphere, 2025_

## Referee Comment (RC1)

A review of "Regime-dependence when constraining a sea ice model with observations: lessons from a single column perspective" by Molly M. Wieringa and Cecilia M. Bitz

This paper aims to quantify the influence that assimilating different observations of sea ice can have under different sea ice conditions. The authors use a single column sea ice model and use an ensemble data assimilation to test the assimilation of synthetic observations of sea ice concentration, sea ice thickness, radar and lidar freeboard in an idealized experiment. The authors investigate the effects of the assimilation on both the aggregate sea ice concentration, sea ice volume and snow volume, as well as the thickness categories. It was found that the assimilation of sea ice thickness provides the largest, positive impact. Laser freeboard was found to be the second most impactful observation to assimilate, and unlike radar freeboard, it did not degrade the snow volume. For the different regime conditions of sea ice, the study showcases a need to carefully treat the categorized state variables when assimilating observations. The paper is well written and the methods are well described. The figures are well thought out and interpreted thoroughly for the results section.

**Novelty**

The main novelty from the paper is in the assessment of ice thickness and freeboard (both radar freeboard and laser freeboard) and how the assimilation affects different regimes of the sea ice. The novel approach finds a potential best-case scenario for assimilation of different types of sea ice observation. The study will provide a lot of insight for those interested in sea ice data assimilation, and particularly for the assimilation of thickness and freeboard, which is an emerging field of study in sea ice. A key result showed that summer and early autumn assimilation of SIT shows strong skill in reproducing the true sea ice state, summer SIT observations were only first produced a few years ago, and their benefit in data assimilation and sea ice studies has not yet been fully explored. The study also provides insight into comparison of freeboard and thickness assimilation and interestingly shows that freeboard assimilation may be less effective than SIT assimilation. The insight into the effects of the assimilation of different observations on the sub-grid scale categories of the state variables is interesting for those interested in implementing sea ice data assimilation themselves, particularly in the implementation of sub-grid scale thickness assimilation.

**General Comments**

Were any specific criteria chosen to determine locations of the pack, seasonal and first year sea ice i.e. in terms of sea ice concentration or sea ice age (e.g. 0.8 commonly being used to separate pack ice from seasonal ice). Alongside this, a map clearly showing the

locations of the three defined types of sea ice (pack, seasonal, first year), which are described in the first paragraphs of the results section, would be beneficial.

Could the authors provide details of how the nonphysical modeled values in the state vector are post-processed after the assimilation, when they occur?

The discussion (or conclusion) could be further enhanced if the paper outcomes were compared to real world available observations. For example the authors find that summer and late autumn observations are key, and that SIT observations have the most impact. However they do not then reference to Landy et al., 2023 (although it is listed in the references, I could not find it referenced in the paper itself). This could be done for other observations also, which would increase the papers usefulness and insightfulness for others in the field.

**Figures**

For figure 2, the ticks for the top 2 rows of figure and the bottom rows are different, and the labels are only shown on the bottom row, the authors could make the x-axis ticks consistent between all the figures, so that the figure is easier to read.

The authors should add panel labels to the figures and each subfigure, which would allow easier reference to them in the text. On line 211 they refer to figure 4 panel labels but the labels do not exist in the figure.

**Minor Comments**

Line 9: DA is not defined yet.

Line 34-35 add references to SIC assimilation papers

Line 77 add references to mushy-layer and delta-eddington scheme

Line 141-142 confusing wording/sentence – seems like a list is introduced but then not continued.

Line 167 missing word "table"

There are a number of references which appear in the references list which do not seem to be cited in the paper e.g.:

Brennan and Hakim, 2022, Chen et al., 2024, Holland & Kwok, 2012, Landy et al., 2023

---

## Author Comment (AC1)

Regime-dependence when constraining a sea ice model with observations: lessons from a single-column perspective

Manuscript egusphere-2025-2148

The authors would like to thank the editor and both referees for the time and effort that have been dedicated to providing feedback on this manuscript. Please find below, in blue, our responses to referee comments, questions, and concerns. All page and line numbers refer to the revised manuscript.

**Responses to Referee 1**

This paper aims to quantify the influence that assimilating different observations of sea ice can have under different sea ice conditions. The authors use a single column sea ice model and use an ensemble data assimilation to test the assimilation of synthetic observations of sea ice concentration, sea ice thickness, radar and lidar freeboard in an idealized experiment. The authors investigate the effects of the assimilation on both the aggregate sea ice concentration, sea ice volume and snow volume, as well as the thickness categories. It was found that the assimilation of sea ice thickness provides the largest, positive impact. Laser freeboard was found to be the second most impactful observation to assimilate, and unlike radar freeboard, it did not degrade the snow volume. For the different regime conditions of sea ice, the study showcases a need to carefully treat the categorized state variables when assimilating observations. The paper is well written and the methods are well described. The figures are well thought out and interpreted thoroughly for the results section.

**Novelty**

The main novelty from the paper is in the assessment of ice thickness and freeboard (both radar freeboard and laser freeboard) and how the assimilation affects different regimes of the sea ice. The novel approach finds a potential best-case scenario for assimilation of different types of sea ice observation. The study will provide a lot of insight for those interested in sea ice data assimilation, and particularly for the assimilation of thickness and freeboard, which is an emerging field of study in sea ice. A key result showed that

summer and early autumn assimilation of SIT shows strong skill in reproducing the true sea ice state, summer SIT observations were only first produced a few years ago, and their benefit in data assimilation and sea ice studies has not yet been fully explored. The study also provides insight into comparison of freeboard and thickness assimilation and interestingly shows that freeboard assimilation may be less effective than SIT assimilation. The insight into the effects of the assimilation of different observations on the sub-grid scale categories of the state variables is interesting for those interested in implementing sea ice data assimilation themselves, particularly in the implementation of sub-grid scale thickness assimilation

We thank the reviewer for highlighting the novelty and timeliness of this work, and for their kind words regarding the manuscript's structure and writing.

**General Comments**

Were any specific criteria chosen to determine locations of the pack, seasonal and first year sea ice i.e. in terms of sea ice concentration or sea ice age (e.g. 0.8 commonly being used to separate pack ice from seasonal ice). Alongside this, a map clearly showing the locations of the three defined types of sea ice (pack, seasonal, first year), which are described in the first paragraphs of the results section, would be beneficial.

We determined the selected locations based on the annual cycle of sea ice concentration. For PACK ICE, we required that the ensemble mean SIC remained above 0.8 throughout the year. For SEASONAL ICE, the criteria were an ensemble that experienced ensemble mean SIC greater than 0.8 and less than 0.15 during the annual cycle, and for FIRST-YEAR ICE, we required a location where all ensemble members melted completely for at least some portion of the year. As this comment is aligned with a similar comment from the second reviewer, we have added these selection criteria to the methods (*Prior Ensembles*, L159-164). The requested map figure has also been introduced as Figure 1 (*referenced at* L169-170).

Could the authors provide details of how the nonphysical modeled values in the state vector are post-processed after the assimilation, when they occur?

We have added a description of the postprocessing approach to the methods section (*DART*, L125-141). If the reviewer is interested in the more specific details of the postprocessing

algorithm, it is also available on Github as a part of the official DART release (https://github.com/NCAR/DART/blob/main/models/cice/ice\_postprocessing\_mod.f90).

The discussion (or conclusion) could be further enhanced if the paper outcomes were compared to real world available observations. For example, the authors find that summer and late autumn observations are key, and that SIT observations have the most impact. However, they do not then reference to Landy et al., 2023 (although it is listed in the references, I could not find it referenced in the paper itself). This could be done for other observations also, which would increase the papers usefulness and insightfulness for others in the field.

We assume that this comment refers to a need to provide more real-world context as to the availability of different types of observations in ice conditions and times of year in which our results indicate they would be most useful. We agree that this would strengthen the manuscript and have added text to the discussion section of the paper that expands upon this point. L376-392 in *Discussions and Implications* now reads:

"Across the three sea ice regimes, assimilating observations during the summer and early autumn freeze-up period demonstrated the greatest ability to reconstruct the true sea ice state. Of the observation types tested, SIT observations had the broadest, and often the largest, positive impact. Not only is MAE for modeled SIT reduced in all regimes, but assimilating SIT also reduces modeled SIC error in seasonally ice-covered regimes. This result arrives at a promising time, as advances in observation processing increasingly enable year-round estimates of SIT from remote sensing observations (Landy et al., 2022). While real, basin-wide SIT observations are still relatively limited during the summer months, these findings should reinforce efforts to improve SIT observing systems during the summer, particularly preceding the transition to freezing conditions.

A novel finding of this study is the comparative impact of FBR and FBL observations, which differ in their impact on modeled snow depth in thick ice environments. While these two types of freeboard observations both appear to constrain thin ice environments very well, it must be noted that for present-day observing systems, both FBR and FBL observations

are only available in regions that exceed a relatively high sea ice coverage threshold (Petty et al., 2023; Kurtz, Galin & Studinger, 2014). Thus, FBL observations are likely to have a more positive overall impact compared to FBR observations, as FBL may also improve SIC and avoid degrading SND in thick ice regimes. Additionally, while freeboard observations of either type display very limited impact in seasonal ice, SIC observations perform comparably to SIT observations in this regime. Given their year-round availability and relatively low observational uncertainty compared to current SIT observational estimates, assimilating a year-round combination of FBL and SIC observations is likely to produce the most accurate sea ice state estimate in current three-dimensional applications."

We assume that the reviewer refers to Landy et al. (2022): A year-round satellite sea-ice thickness record from Cryosat-2, which was published in Nature. We have included this reference explicitly (above) and made sure it is listed properly in the references section.

**Figures**

For figure 2, the ticks for the top 2 rows of figure and the bottom rows are different, and the labels are only shown on the bottom row, the authors could make the x-axis ticks consistent between all the figures, so that the figure is easier to read.

Thanks to the reviewer for catching this mistake! It has been corrected in the figure, which is now Figure 3 in the updated manuscript.

The authors should add panel labels to the figures and each subfigure, which would allow easier reference to them in the text. On line 211 they refer to figure 4 panel labels but the labels do not exist in the figure.

We appreciate this suggestion and have added figure labels to multi-panel figures that are referenced in the text (Figures 2-6, Figure 14 in the updated manuscript).

**Minor Comments**

Line 9: DA is not defined yet.

Thanks for catching this! We have addressed this issue by defining data assimilation (DA) in line 2.

Line 34-35 add references to SIC assimilation papers

We have added references to Lisaeter et al. (2003), Schweiger et al. (2011), and Posey et al. (2015) as examples for SIC assimilation efforts at L35.

Line 77 add references to mushy-layer and delta-Eddington scheme

The relevant references (Turner, Hunke, & Bitz, 2013; Briegleb & Light, 2007; Holland et al., 2012) have been added to the manuscript at L77-78.

Line 141-142 confusing wording/sentence – seems like a list is introduced but then not continued.

This was a missing component of the reference to Table 1. It has been corrected at L157 in the revised manuscript.

Line 167 missing word "table"

Thanks for catching this omission, as well! We have corrected it (L189 in the revised manuscript).

There are a number of references which appear in the references list which do not seem to be cited in the paper e.g.: Brennan and Hakim, 2022, Chen et al., 2024, Holland & Kwok, 2012, Landy et al., 2023

We would like to express our appreciation for the reviewer's attention to detail in the reference section—we would likely have missed these errors without this comment! These citations have been removed from the references section, which has also been thoroughly checked for any other inconsistencies.

---

## Author Comment (AC2)

Regime-dependence when constraining a sea ice model with observations: lessons from a single-column perspective

Manuscript egusphere-2025-2148

The authors would like to thank the editor and both referees for the time and effort that have been dedicated to providing feedback on this manuscript. Please find below, in blue, our responses to referee comments, questions, and concerns. All page and line numbers refer to the revised manuscript.

**Responses to Referee 2**

The manuscript entitled "Regime-dependence when constraining a sea ice model with observations: lessons from a single-column perspective" investigates the impact of several plausible observation types of sea ice on the sea ice variables in three different sea ice regimes: PACK ICE, SEASONAL ICE, and FIRST-YEAR ICE. The authors conducted a series of perfect model experiments using the linked DART and Icepack and analyzed the results using several metrics comprehensively. The authors find that the efficacy of sea ice DA varies significantly across observation types and sea ice regimes. While the former finding is not new, the latter finding is novel. This study suggests that DA strategies need to be tailored to observation types and regimes, which provides insights for the sea ice DA community. The experiments are well-designed, and the manuscript is well written. I very much enjoyed reading it. Albeit, I have several comments for the authors to address before publication, which I consider minor.

We thank the reviewer for nicely summarizing the contributions of the manuscript and we are very glad to hear that it was enjoyable to read!

1. The authors presented a lot of results in the paper, yet do not provide enough speculations or discussions. It's helpful to know which observation types perform better than others, or which observations work better in certain regimes, but are they model specific? How translatable are the findings? Digging deeper into the mechanisms that contribute to their different performance might be more valuable.

We appreciate the reviewer's request for further discussion and speculation regarding the broader impact of these findings.

The perfect-model experiments are useful for understanding the DA process under optimal conditions. They disregard several important considerations for real-world applications, including but not limited to observation representativeness, observation availability and rejection, intractability of observational error structures, covariance localization, and model bias compared to the real world. As such, one should expect that the part of the findings that relate to DA "mechanics" should be consistent, and parts that are functions of the model or the observations themselves may need to be treated with care when scaling up or complicating our experimental approach. For example, the model covariance relationships between variables are model-dependent but also tend to reflect common tendencies or difficulties across models (e.g. the tight spread in SIC in winter, since freezing conditions lead to total sea ice coverage in a most grid cells, versus the broader spread in SIT/FBR/FBL). We have added the following sentence at L402-404.

"While the details of these covariance relationships are dependent on how the model represents variability in the sea ice state, their general structure and seasonal evolution reflect physics that should be consistent across credible sea ice models available today."

In terms of observations, the error structures used in the work are modeled on those from current instruments or previous literature but are not fixed for all observation types and may influence the impact of a specific observational type. Real-world observations are also not likely to share the one-to-one representativity used in this work, which may alter their impact. We have made several references to these kinds of caveats throughout the paper (L143-149, L186-188, L368-373, L393-404) and clarified that the results should be considered an upper bound on the efficacy of these observation types (L58-60, L147-149, L373-374).

In several instances, discussion of why an observation type has a particular impact is presented immediately with the results pertaining to that observation. We made this choice to avoid referring to many different pieces of the work several times across sections, though we acknowledge that it makes the *Results* section noticeably longer than the *Discussion & Implications* section. The manuscript presents several relevant mechanisms for observation performance across regimes, including seasonal model ensemble spread compared to observational error magnitude (L300-316, L358-361), covariance relationships between

observation quantity and state variables (L393-404), proximity of the selected TRUTH to the FREE ensemble mean (L289-290) and observational error structure (L405-414).

We have expanded some discussion and speculation regarding which types of observations are likely to produce the best estimate of the state in present-day applications, given observational availability, in L379-392.

2. The format of the paper needs to be cleared up. Figures are referred to as Fig. XX in some places and Figure. XX elsewhere. Please be consistent throughout the paper. Also, there are no sub-labels in the plots, some of which are fine, but not when they are referred to in the paper, e.g., Figure 4. The citation also needs to be edited, e.g., Petty et al (2023a) is referred to in the text, but it's not clearly labeled in the reference list section

We thank the reviewer for catching these errors. For clarification, we are using The Cryosphere submission requirements, which are to use the abbreviation "Fig. XX" when referencing figures, except at the beginning of a sentence. We have checked the manuscript for consistency with respect to this stylistic choice. We have also added panel labels to multi-panels figures (Figs. 2-6, Fig. 14). Finally, in conjunction with similar notes from another reviewer, we have thoroughly checked the references section and citations for consistency. The particular note regarding Petty et al (2023a) has been corrected by removing the "a" designation, since there is no longer a Petty et al (2023b) in the references (L43 in the revised manuscript).

More detailed comments are listed below

1. The definition of the three regimes is a bit sporadic in the paper. Since it's an important concept in the paper, I'd suggest clearly defining them in the experimental setup, maybe in a separate subsection.

In conjunction with a similar comment from another reviewer, the selection criteria and definitions for the regimes have been added to the methods section (*Prior Ensembles*, L159-164). A map detailing the specific locations being modeled by each regime ensemble has also been added as Figure 1 (*referenced at* L169-170).

2. There's a lot of information in Figure 2 that's not discussed in the paper. For example, although the aggregate SIC spread is small throughout the year except in the summer, the

spreads of individual categories are decent. What does this suggest, and how will this impact DA results? I believe these discussions are valuable.

We are grateful to the reviewer for highlighting some shortcomings when it comes to how we have presented Figure 3 (previously Figure 2). In reviewing this comment, it became apparent to us how a lack of direct reference to Figure 3 in the initial results section makes it difficult to link the text to the figure. We have added end-of-sentence references to specific panels of Figure 3 when examining how Icepack ensembles represent each regime (L212-214, L217-218, L225, L228). It is our hope that this more explicitly links the discussion of spread and mean state in the ensembles in this section to the relevant parts of Figure 3.

Regarding the reviewer's specific comment on the relationship between spread in aggregate SIC and spread in individual  $A_{ice,n}$  categories in the freezing season, we note that this is possible since there are many ways that 5 categories can be distributed but still add up to 1. There are two primary implications. First, a DA adjustment could move ice around the categories while not appearing to shift the total concentration very much; this is expected behavior in winter when one expects SIC in PACK ICE regimes to be close to 1. Secondly, however, adjustment in each of the categories is dependent upon how much an observation can adjust the model's estimate of that observed quantity and the strength of the covariance relationship between observable and state quantities in the model (Fig. 14 in the revised manuscript). As an example, since spread in SIC is so small in the PACK ICE ensemble, assimilating an SIC observation is unlikely to produce a substantial adjustment to the model's ensemble mean estimate of SIC (the DA sees the ensemble as a "very certain" estimate). If there isn't a substantial update to SIC in the model ensemble, then there won't be much of an increment to regress onto each of the individual categories when updating the state variables (even less so if the covariance relationship used to perform the regression is also weak). This is part of why SIC observations should not be expected to be effective when SIC spread is low, even if category area spreads are larger; on the other hand, observations like SIT and FBR/FBL (or SIC in thinner ice environments) which have decent ensemble spread in the winter, could produce a substantial increment that could be regressed to accurately update the categories during those times (Fig. 11 in the revised manuscript).

While the Figure 3 presentation of the prior ensembles does prelude these ideas without directly introducing them, they are discussed via a few examples from the assimilation experiments in the *Results* section (specifically for SIC, see L335-352). An explicit treatment for how assimilating an aggregate quantity like SIC leads to updates for individual categories like  $A_{ice.n}$  is currently being prepared by the authors for another study—a more detailed discussion of the relationships between aggregate and categorized DA updates will be presented there.

Another thing that catches eyes is the huge spread of SIC in the FIRST-YEAR ICE. Basically it ranges from 0 to close to 1 in winter. Is it representative of the model uncertainty?

The large spread in FIRST-YEAR ICE SIC reflects the variability in freezing conditions in the perturbed ensemble of atmospheric forcings around a mean that is near the melting point even in the winter months (Fig. A1). Given the more southerly location of this grid cell near the ice margin in the Atlantic sector of the Arctic Ocean, it is possible to have atmospheric conditions that stay above freezing and conditions that stay below freezing over the same period in the forcing ensemble. This results in a much wider range of ice coverage conditions than is seen at locations where the atmospheric conditions are well below freezing in all ensemble members for most of the year. We consider this representative of the actual SIC natural variability at marginal ice locations like this one, if not an under-representation (see Fig. 5, panel b). This is lightly discussed in the manuscript (L225-228, L268-271) but has been more explicitly referenced to the figure (now Fig. 3) in the text (L225).

Figure 4 is meant to evaluate if the ensembles in the single column grid points are representative of their regimes, but it only shows the time-averaged (co)variances, which does not consider the large seasonal variation, especially in the FIRST-YEAR ICE regime.

We have adjusted Figure 4 in response to this point (and the more specific comment below) such that the time averaged (co)variances and the comparison on interannual ensemble (co)variances are both presented.

3. The panels in Figure 4 are not labeled but referred to in the text. Please add sublabels.

Panel labels have been added to this figure (now Fig. 5 in the revised manuscript). Thanks for catching this issue.

4. In Figure 4, the right panel doesn't provide additional information to the left panel, at least the authors didn't elaborate on it. I'd suggest re-arrange the panels in Figure 4 to add the seasonal variations of spread and trim the panels that are not discussed in the text.

We consider this suggestion a good one and have made an attempt to address it by shifting the time-averaged (co)variances as a function of mean ice thickness to the left column and adding in the comparison between ensemble (co)variance in each regime and the corresponding grid-cell (co)variance at each day of the year across a five-year CICE6 simulation to the right column. The inclusion highlights the general agreement with respect to timing of variance captured by the Icepack ensembles, but also their underestimation of SIC and SIT variance (as well as their covariance) cycle compared to interannual variability in a CICE6 simulation. A revised and expanded discussion of this figure (now Fig. 5 in the revised manuscript) has been added at L229-243.

5. Block 240: why does FBR differ so much from FBL in FY ICE? The authors discussed their different performances in snow depth estimates but didn't mention their differences in ice volume. The possible contributing factors are also not mentioned. Since the two observation types are two key derived products, it's worth investigating why one offers more value than the other, and the reasons behind it.

The differing impact of FBL and FBR observations for constraining thick and thin ice regimes (in terms of ice thickness) is reviewed in the *Discussion & Implications* section (L408-414). As is mentioned there, the relative superiority of FBL observations in thin ice regimes (FY ICE) is a product of the error structures of each observational type, which have been derived from currently available satellite retrievals of radar and laser freeboard (Fig. B1). In thin ice regimes, error in FBL observations appears small compared to ensemble spread, while FBR error appears large compared to ensemble spread (Figs. 3 & 4). As such, the FBL observations will produce an adjustment, while the FBR observations will not. In thick ice regimes, both observation types have error that is smaller than the ensemble spread, but FBR error appears smaller than FBL error, so FBR observation more tightly constrain SIT in that regime.